# Insights into Aptamer–Drug Delivery Systems against Prostate Cancer

**DOI:** 10.3390/molecules27113446

**Published:** 2022-05-26

**Authors:** Xueni Wang, Qian Zhou, Xiaoning Li, Xia Gan, Peng Liu, Xiaotao Feng, Gang Fang, Yonghong Liu

**Affiliations:** 1Guangxi Zhuang Yao Medicine Center of Engineering and Technology, Guangxi University of Chinese Medicine, 13 Wuhe Road, Qingxiu District, Nanning 530200, China; wangxueni@gxtcmu.edu.cn (X.W.); zhouqian202203@163.com (Q.Z.); ganxiaxxn@163.com (X.G.); 2CAS Key Laboratory of Tropical Marine Bio-Resources and Ecology, Guangdong Key Laboratory of Marine Materia Medica, South China Sea Institute of Oceanology, Chinese Academy of Sciences, Guangzhou 510301, China; 3School of Foreign Languages, Nanning Normal University, 3 Hexing Road, Qingxiu District, Nanning 530299, China; lxnzjl@163.com; 4Institute of Marine Drugs, Guangxi University of Chinese Medicine, 13 Wuhe Road, Qingxiu District, Nanning 530200, China; 5Center for Medical Innovation, School of Basic Medical Science, Guangxi University of Chinese Medicine, 13 Wuhe Road, Qingxiu District, Nanning 530200, China; liup@gxtcmu.edu.cn; 6Guangxi Key Laboratory of Chinese Medicine Foundation Research, Guangxi University of Chinese Medicine, 13 Wuhe Road, Qingxiu District, Nanning 530200, China; fengxt2008@163.com

**Keywords:** aptamer, drug delivery systems, prostate cancer, doxorubicin, paclitaxel, docetaxel, curcumin, thymoquinone

## Abstract

Prostate cancer is a common cancer in elderly males. Significant progress has been made in the drug therapies for prostate cancer in recent years. However, side effects are still problems that have not been overcome by the currently used anti-prostate cancer drugs. Novel technologies can be applied to reduce or even eliminate the side effects of drugs. An aptamer may be a sequence of nucleic acids or peptides that can specifically recognize proteins or cells. Taking advantage of this feature, scientists have designed aptamer–drug delivery systems for the development of anti-prostate cancer agents. Theoretically, these aptamer–drug delivery systems can specifically recognize prostate cancer cells and then induce cell death without attacking normal cells. We collected the relevant literature in this field and found that at least nine compounds have been prepared as aptamer–drug delivery systems to evaluate their precise anti-prostate cancer effects. However, the currently studied aptamer–drug delivery systems have not yet entered the market due to defects. Here, we analyze the published data, summarize the characteristics of these delivery systems, and propose ways to promote their application, thus promoting the development of the aptamer–drug delivery systems against prostate cancer.

## 1. Introduction

Prostate cancer is the fifth leading global cause of death in men [1]. At present, the drug therapies for prostate cancer are mainly endocrine therapy and chemotherapy. The principle of endocrine therapy for prostate cancer is to eliminate the stimulating effect of androgen on prostate cancer cells. These therapies can be achieved by drugs such as GnRH-agonists or antagonists and non-steroidal antiandrogens [2]. However, these therapies have many shortcomings, including nonspecific toxicity, multidrug resistance, and the recurrence rate. It is hard to achieve a satisfactory prognosis for patients who receive these therapies. Scientists have looked for more efficient and safer therapies. Several targeted drug delivery systems have been applied for enhancing the anti-tumor efficacy of chemotherapeutics, including liposomes, solid lipid nanoparticles, micelles, gold nanoparticles, polymeric nanoparticles, carbon nanotubes, carbon dots, iron oxide nanoparticles, and dendrimers [3]. In addition, many precise targeting technologies have been developed for the treatment of prostate cancer, including vaccines, antibody–drug conjugates (ADCs), and aptamer–drug conjugates (ApDCs).

Therapeutic cancer vaccines have been recognized as a promising treatment option in clinical oncology for prostate cancer. However, only one cancer vaccine (sipuleucel-T) has obtained Food and Drug Administration (FDA) approval for clinical use to date. ADCs are another drug delivery system that has rapidly developed in recent years. To date, many ADCs have been subjected to clinical trials around the world. Although dozens of ADCs have entered the market, none of the approved ADCs were designed for the treatment of prostate cancer [4,5]. The ADC consists of an antibody moiety, which ensures its high specificity to the target and high target-binding affinity, along with low immunogenicity and low cross-reactivity. Prostate-specific membrane antigen (PSMA) is a transmembrane glycoprotein that is overexpressed in malignant prostate tumor cells [6]. PSMA has been considered as a target for the diagnosis and treatment of prostate cancer. PSMA-targeted ADCs have shown anti-tumor activity in prostate cancer; however, treatment-related adverse events temporarily limit its application [7]. ApDC is another type of drug delivery system that can accurately identify specific proteins or specific cells. With the progress of aptamer screening technology, an increasing number of aptamers that can directly identify specific cells have been found. The aptamer–drug delivery system can directly distinguish target cells from non-target cells, and this technology greatly reduces the side effects of drugs. The aptamer–drug delivery system has become a popular topic in the field of precise drug delivery.

Aptamers are small nucleotide sequences or amino acid sequences that can comprise RNA, single-stranded DNA oligonucleotides, DNA–RNA hybrid oligonucleotides, peptides, etc. These sequences have high specificity and affinity for certain cells or proteins. Aptamers have more advantages compared to antibodies; for instance, aptamers have a smaller size, higher binding affinity, higher specificity, better biocompatibility, higher stability, and lower immunogenicity, which all contribute to their wide application in the biomedical field [8,9,10,11]. In addition, antibodies can only identify immunogenic targets, but aptamers can widely identify ions, polypeptides, small-molecule compounds, proteins, nucleic acids, viruses, bacteria, cells, and tissues [12]. Moreover, the production cost of aptamers is lower than that of antibodies [13]. Many aptamers can be selected and generated from large synthetic libraries through the systematic evolution of ligands by exponential enrichment (SELEX) technology [14,15,16,17,18].

Here, we collected the literature related to aptamer–drug delivery systems for the treatment of prostate cancer. First, we collected compounds that were used to prepare aptamer–drug delivery systems for treating prostate cancer. Then, we classified these aptamer–drug delivery systems according to the type of preparation, and summarized the effect and clinical value of these agents. An aptamer–drug delivery system is an effective targeted drug delivery system, and is constructed by combining aptamer sequences with therapeutic drugs by chemical modification or drug carriers. This combined treatment method can enhance cytotoxicity, reduce IC_50_, and improve the therapeutic effect on tumor cells, thus reducing the damage to normal cells and tissues. The aptamer–drug delivery system is expected to become a new and improved drug delivery system following the emergence of ADCs.

## 2. Aptamer–Drug Delivery Systems against Prostate Cancer

### 2.1. Aptamer–Doxorubicin Drug Delivery Systems

Doxorubicin (Dox) is one of the most widely used anthracycline anti-cancer drugs for the treatment of solid tumors (such as prostate cancer, ovarian cancer, breast cancer, and gastrointestinal cancer). However, these anti-cancer drugs have various side effects, such as allergic reactions, heart injuries, hair loss, bone marrow suppression, vomiting, and bladder stimulation, which limit their clinical application [19]. In order to reduce or even eliminate the toxicity and side effects of Dox, scientists have undertaken a series of efforts. One strategy is to combine Dox with an aptamer, and then combine them in various pharmaceutical preparations.

Boyacioglu et al. [20] proposed a new dimeric aptamer complex called DAC. DAC has a good binding affinity and therapeutic efficacy as a dimeric aptamer complex [21]. DAC can specifically target PSMA. Boyacioglu et al. [20] developed a complex DAC-Dox (DAC-D) with stoichiometry of ~4:1 by covalent modification of DAC with Dox. The covalent bond in DAC-D is sensitive to pH, which makes DAC-D release Dox in acidic environments, internalize it into target cells, and exert a drug effect. DAC-D has a low molecular weight, high permeability, and long residence time, and can efficiently target tumor cells, and enhance drug delivery and drug efficacy. It is well known that C4-2 cells express PSMA, whereas PC-3 cells do not express this protein. In this study, DAC-D displayed more than 80% inhibition of targeted C4-2 cells and less than 50% cytotoxic activity towards PC-3 cells. DAC-D showed specificity and stability characteristics, which may help to improve the selective delivery of Dox to malignant tissues in vivo and provide a new therapy for prostate cancer.

DUP-1 is a peptide aptamer targeting PSMA-negative cells, and A10-3.2 is an RNA aptamer targeting PSMA-positive prostate cancer cells. Jing et al. [22] designed a dual-aptamer modified tumor-targeting gene and Dox delivery system mediated by recombinant adenovirus (A10-3.2(Dox)/DUP-1-PEG-Ad5, ADDP-Ad5). They found that LNCaP and PC-3 prostate cancer cells became round and sparse after being treated with ADDP-Ad5, the gap between cells was larger, and cells showed obvious damage. These findings suggested that ADDP-Ad5 significantly inhibited the growth of LNCaP and PC-3 cells. At the same time, they found that the uptake rate of ADDP-Ad5 was 90.35 ± 1.22% and 78.71 ± 2.53% for LNCaP and PC-3 cells, respectively. On the contrary, they found ADDP-Ad5 showed little or no inhibition on the cell growth of human normal prostate epithelial cells. Furthermore, the in vivo antitumor activity test showed that ADDP-Ad5 was active against LNCaP and PC-3 tumor xenografts in vivo and had no obvious toxicity to mice.

A9 is an aptamer that is a well-known PSMA-specific RNA sequence. Kim and Lee et al. [23,24] designed an aptamosome that comprised an aptamer-conjugated liposome encapsulating A9, vimentin siRNA, and Dox. They proved that the aptamosome was significantly more cytotoxic toward targeted LNCaP cells than the non-targeted PC-3 and A549 cells, suggesting that the aptamosome improved the targeting ability of Dox.

A10 is a commercially available aptamer targeting PSMA. Xu et al. [25] created a A10 aptamer-conjugated unimolecular micelle loaded with Dox, which exhibited a pH-sensitive and controlled drug release behavior. The experimental results showed that Dox-loaded targeted single-molecule micelles exhibited approximately 50% growth inhibition at a dose of 1 mg/mL. Untargeted single-molecule micelles showed only about 24% inhibition at the 1 mg/mL dose of Dox. The targeted unimolecular micelles exhibited a much higher cellular uptake in PSMA-positive CWR22Rv1 prostate carcinoma cells than non-targeted unimolecular micelles, thereby leading to a significantly higher cytotoxicity. These findings suggested that aptamer-conjugated unimolecular micelles may potentially be an effective drug nanocarrier to effectively treat prostate cancer. In another study, an A10 aptamer and Dox were used to construct a polylactide nanoconjugate named A10 Dox-PLA NCs. By virtue of controlled drug release kinetics and selective tumor-associated endothelial cell targeting, A10 Dox-PLA NCs possess a desirable safety profile in vivo, being well tolerated following high-dose intravenous infusion in mice, as supported by the absence of any histologic organ toxicity [26]. Quantum dots (QDs) are engineered fluorescent nanoparticles with unique optical and chemical properties, and have shown great potential as promising platforms for biomedical applications [27]. Semiconductor nanocrystalline quantum dots have been increasingly designed to carry different types of therapeutic agents for disease treatment applications due to their unique optical properties, including wide absorption, narrow luminescence spectrum, high quantum yield, low photobleaching, and chemical degradation resistance. In Bagalkot’s study [28], a new quantum dot (QD)-A10-Doxorubicin was assembled and used to combat prostate cancer. The study found that LNCaP cells effectively absorbed QD apt conjugates, whereas PC-3 cells bound less to the conjugates. Cell proliferation testing showed that, although the cytotoxicity of free Dox to LNCaP and PC-3 cells was equal, the cytotoxicity of the QD-A10-Dox delivery system against LNCaP cells was significantly higher than that of the non-targeted PC-3 cells (cellular viability: LNCaP 52.5 ± 1.6% versus PC-3 77.2 ± 3.1%; mean ± SE, *n* = 3; probability value *p* < 0.005).

Apt 1 is a ssDNA aptamer that was selected to detect LNCaP cells. Atabi et al. [29] used glutaraldehyde to link the aptamer with myristilated chitosan nanogels (MCS), then loaded them with Dox to construct a targeted drug delivery system (Apt-MCS-Dox). Apt-MCS-Dox specifically bound to LNCaP cells, whereas it did not show any specificity to PC-3 cells, which were set as a negative control. MTT assay was performed to test the cytotoxicity of the targeted drug delivery system on LNCaP cells. The results indicated that the application of MCS-DOX and Apt-MCS-DOX showed significant (*p* < 0.001) lethal effects on LNCaP cell lines.

A9-(CGA)_7_ is a PSMA-specific RNA aptamer that is an extended version of the A9 aptamer. Nanoparticles have a low molecular weight and readily promote drug absorption. Kim et al. [30] first linked an A9-(CGA)_7_ aptamer with gold nanoparticles, and then loaded Dox into the complex to form aptamer-gold nanoparticles-Dox. They found that the aptamer-Dox-GNPs can bind specifically to LNCaP prostate cancer cells, and showed more significant cytotoxicity on LNCaP cells than the PC-3 cells, which do not express PSMA (cell viability 50 ± 6% for LNCaP versus 71 ± 6% for PC-3; *p* < 0.05, *n* = 5).

A new RNA–DNA chimeric hybridization aptamer, Apt·dONT-DEN, was designed using double strands of a DNA-A9-(CGA)_7_ aptamer and a dendrimer to target PSMA [31]. Subsequently, Apt·dONT-DEN was applied to combine with Dox to form a drug delivery system (Dox@ Apt·dONT-DEN) against prostate cancer. The Dox@ Apt·dONT-DEN system showed excellent antitumor effects in vitro and in vivo. The Apt·dONT-DEN vector showed no obvious cytotoxicity to prostate cancer cells (LNCaP, 22Rv1, DU145, and PC-3 cells), but could deliver Dox to target cells in a specific way, showing its superiority in the treatment of prostate cancer.

An unnamed aptamer targeting PSMA was used to form a shRNA/PEI-PEG-APT/Dox delivery system in the study of Kim et al. [32]. Cell viability assays showed that the conjugates inhibited the growth of PSMA-abundant prostate cancer cells with strong cell selectivity. In addition, IC_50_ values of their aptamer-based delivery system were approximately 17-fold lower than those for the simple mixture of shRNA plus the drug. It was shown that the aptamer improved the targeting of drugs.

AS1411 is an aptamer targeting nucleolin. It can be used to deliver a wide range of drugs to cancer cells [33]. Taghdisi et al. [34] designed a novel Dox-loaded three-way junction pocket DNA nanostructure containing three strands of AS1411 aptamer to treat PC-3 cells. The PC-3 cell line is a prostate cancer cell line that expresses nucleolin. After being treated with Dox, DNA nanostructure, Dox-loaded control complex, and Dox-loaded DNA nanostructure, the cell viability of PC-3 cells was 46.8 ± 2.7%, 85.7 ± 9%, 80.1 ± 3.5%, and 33.2 ± 1.5%, respectively. Cell viability assay demonstrated that the Dox-loaded DNA nanostructure had significantly higher cytotoxicity for PC-3 cells compared to non-target cells. Furthermore, the in vivo experiment showed that the Dox-loaded DNA nanostructure was more effective in prohibition of the tumor growth compared to free Dox.

Ecad01 is a novel aptamer targeting E-cadherin. It was used to synthesize Ecad01-Dox conjugates for target-specific delivery of doxorubicin (Dox) to inhibit prostate cancer cell (DU145) proliferation. The Ecad01-Dox conjugates exhibited excellent targeted internalization, which was evident from a 1.71-fold increase in internalization in DU145 cells, and showed a significantly lower uptake (1.92-fold lower) in non-cancerous cells (RWPE-1). Moreover, cell viability assay results showed that 1.0 μM Dox in the Ecad01-Dox conjugates was able to show similar cytotoxicity to 10 μM Dox in DU145 cells, which is indicative of specific targeted cancer inhibition. This study proved that the Ecad01-Dox conjugates can specifically kill prostate cancer cells and reduce the survival rate of DU145 cells. Moreover, the Ecad01-Dox conjugates do not have any obvious toxic effect on healthy cells, can reduce the nonspecific cytotoxicity of the conjugates in normal cells, and can increase the internalization potential and efficacy [35].

A10-3-J1 is a DNA-RNA hybrid aptamer invented by Leach et al. [36] to target PSMA. A10-3-J1 and Dox were combined with a super paramagnetic iron oxide to form the nanoparticle aptamer Doxorubicin (A10-3-J1-SPIO-APT-DOX). The study determined the potential of A10-3-J1-SPIO-APT-DOX as a therapeutic drug for prostate cancer by evaluating its effects on LNCaP cells and PC-3 cells. They found the agent can specifically recognize PSMA (+) prostate cancer cells, enhance the cytotoxicity, promote the endocytosis of target cells, and minimize the collateral damage to non-target cells. In addition, it can significantly reduce or even eliminate the adverse reactions of Dox. Treatment of LNCaP cells with Dox loaded with A10-3-J1-SPIO NPs resulted in twice the cell death, whereas most PC-3 cells survived after treatment.

A10/DUP-1 is a new dual-aptamer system that was developed to target both PSMA positive and negative prostate cancer cells. Superparamagnetic iron oxide nanoparticles (SPION) have gradually gained attention due to their low toxicity, stronger proton relaxation, and lower detection limit. Scientists immobilized both aptamers onto the thermally cross-linked superparamagnetic iron oxide nanoparticles (TCL-SPIONs) to develop the dual-aptamer-based drug delivery system. Then, Dox was loaded with the A10/DUP-1-TCL-SPION to obtain a novel target drug delivery system. They found that this system can induce the death of both types of prostate cancer cells through the process of apoptosis [37,38].

In summary, the aptamers used in Doxorubicin are mainly the RNA aptamer, DNA aptamer, and RNA-DNA hybridization aptamer (Table 1). Most of the aptamers target the PSMA protein, whereas one targets the E-cadherin protein and one targets the nucleolin protein. The cells used in experiments were mainly LNCaP cells, 22Rv1 cells, and C42B cells, which express PSMA. Nanoparticles are the most used preparations, in addition to myristilated chitosan nanogels, liposomes, and micelles.

### 2.2. Aptamer–Cisplatin Delivery Systems

Cisplatin is one of the most effective chemotherapeutic drugs for the treatment of many types of solid tumor [39]. Although it is not the first-line drug for prostate cancer chemotherapy [40], the development of cisplatin offers new hope for the treatment of prostate cancer [41]. Clinically, the application of cisplatin is limited because of its inherent and acquired drug resistance to prostate tumor cells, and other toxic and side effects [42]. In the experiments of Shanta Dhar et al. [43], the hydrophobic Pt (IV) compound 1 was used as a prodrug for delivering cisplatin to prostate cancer by PSMA-targeted PLGA-b-PEG-NPs, which were made of poly(D,L-lactic-co-glycolic acid)-b-poly(ethylene glycol) (PLGA-b-PEG) nanoparticles (NPs). The PLGA-b-PEG-Pt (IV) prodrug NPs increased the drug’s maximal tolerated dose when compared to that of cisplatin administered by its conventional dosage form in animal experiments. Making hydrophobic Pt (IV) compound 1 into a prodrug targeted delivery system of cisplatin can achieve safer and more effective prostate cancer therapy in vivo. In the work of Har et al. [44], the authors used the A10 aptamer and poly(D,L-lactic-co-glycolic acid (PLGA)-b-poly (ethylene glycol) (PEG) (PLGA-b-PEG) to develop nanoparticles to specifically deliver cisplatin to prostate cancer cells that express PSMA. Then, they adopted a new route of intravenous administration and detected the cytotoxicity of LNCaP cells. They found that the toxicity of platinum (IV)-encapsulated PLGA-b-PEG nanoparticles targeted by the A10 aptamer was 80 times higher than that of free cisplatin in LNCaP cells. The aptamer–drug delivery systems greatly improved the efficacy of cisplatin and reduced its cytotoxicity to non-targeted cells. It is worth noting that, in addition to cisplatin and platinum-based drugs, ruthenium and other metal complexes have been applied to directly (by covalent conjugation) or indirectly (via the use of a suitable platform) link to specific aptamers to form novel drug delivery systems [45]. These studies will promote the development of aptamer–metal complex drug delivery systems against prostate cancer.

### 2.3. Aptamer–Curcumin Delivery Systems

Curcumin (Cur) is a phenolic compound that exerts a wide range of beneficial effects, including anti-microbial, anti-oxidant, anti-inflammatory, and anti-cancer effects [46,47]. However, the clinical application of curcumin is limited by its low absorption rate, rapid metabolism, and poor bioavailability [47,48]. Liposomes can effectively improve the stability of drugs and prolong their storage time. The aptamer A15 has been proven to be a promising ligand for targeting CD44^+^/CD133^+^ cells [49,50]. Taking advantage of the liposomes and aptamer A15, Ma et al. [51] designed A15 curcumin liposomes (A15-Cur LPs) by modifying curcumin liposomes with the aptamer A15 for targeting prostate cancer stem cells (CSCs). It was found that the drug–aptamer complex can specifically recognize the cells expressing CD44^+^/CD133^+^ on the surface of prostate cancer cells, and inhibit the growth and induce apoptosis of prostate stem cells by prolonging circulation, and increasing permeability and retention (EPR) effects. In an experiment involving a solid tumor transplanted with DU145 cells in mice, it was found that the tumor volume of mice can be significantly reduced using A15-Cur-LPs.

### 2.4. Aptamer–Docetaxel Delivery Systems

Docetaxel (Dtx) is an M-phase cycle specific drug that has cytotoxic effects on a variety of human tumor cell lines [52]. Prostate cancer is one of its clinical indications [53]. However, its clinical application is limited by its dose limiting toxicity, poor solubility, serious side effects, and multi-drug resistance (MDR) [54]. Therefore, scientists have explored the administration route of Dtx. Farokhzad et al. [55] constructed docetaxel (Dtx)-encapsulated nanoparticles formulated with a biocompatible and biodegradable poly(D, L-lactic-co-glycolic acid)-block-poly (ethylene glycol) (PLGA-b-PEG) copolymer and surface functionalized with the A10 2′-fluoropyrimidine RNA aptamers that recognize the extracellular domain of the PSMA. Experiments showed that the combination of the aptamer and drug improved the targeted delivery and uptake of drugs. Chen et al. [56] also studied aptamer nanoparticle Dtx-apt-NPs and tested their antitumor effect in vivo on a LNCaP cell xenograft tumor model. They found that the new complex can be absorbed not only through membrane mobile transport, but also through active transport (aptamer mediated endocytosis). In addition, the complex Dtx-apt nanoparticles can more effectively induce LNCaP cell apoptosis or death through G2/M phase cell cycle arrest compared to Dtx-free nanoparticles.

### 2.5. Aptamer–Monomethyl Auristatin Conjugates

Monomethyl auristatin E (MMAE) is a potent anti-cancer microtubule-targeting agent (MTA) [57]. Monomethyl auristatin F (MMAF) is a new antimitotic auristatin derivative with a charged C-terminal phenylalanine residue that attenuates its cytotoxic activity compared to its uncharged counterpart, MMAE, most likely due to impaired intracellular access [58]. Both auristatin drugs are potent microtubule inhibitors that are too toxic to be used in an unconjugated state and need to be linked to a monoclonal antibody [59]. However, these drugs are very suitable for effective tumor-targeted therapy. E3 is a new prostate cancer-specific RNA aptamer that can specifically recognize PSMA, which is determined by cell selection technology [59]. E3 aptamers have been coupled to highly toxic chemotherapeutic drugs and have been shown to target and treat prostate tumors. Aptamer highly toxic drug conjugates (ApTDCs) based on E3 aptamers were selected on prostate cancer cells to target and inhibit the growth of prostate tumors in vivo. Gray et al. [60] found that ApTDCs formed by coupling E3 with the drugs monomethyl auristatin E (MMAE) or monomethyl auristatin F *(*(MMAF) can effectively target and kill a series of different cancer cells. Confocal microscopy experiments confirmed that E3 was internalized into prostate cancer cells through active receptor-mediated endocytosis, resulting in accumulation in lysosomes. Both MMAE-E3 and MMAF-E3 drug conjugates can effectively kill prostate cancer cells without affecting the vitality of normal prostate cells. The significance is that the E3 aptamer is located in the mouse prostate xenograft. MMAF-E3 significantly inhibits the growth of the mouse prostate tumor and prolongs the survival time of mice, which proves the utility of aptamer–drug conjugates in vivo. One advantage of ApTDCs is that they can be easily controlled and quickly inactivated by antisense oligonucleotide “antidotes”. The anti-E3 antidote has been shown to prevent E3 cell targeting and MMAE-E3 and MMAF drug coupling toxicity [59].

### 2.6. Aptamer–Paclitaxel Delivery Systems

Paclitaxel (Ptx), a tetracyclic diterpenoid compound, was first isolated from the bark of the Pacific yew tree. Paclitaxel has been widely applied to combat ovarian cancer, breast cancer, uterine cancer, and other cancers because it is a high efficiency and broad-spectrum natural anti-cancer drug [61]. Ptx is the first-line chemotherapeutic agent for patients with castration-resistant prostate cancer [62]. However, its current clinical utility has been limited due to numerous serious side effects and drug resistance [63]. Therefore, there is an urgent need to develop an effective treatment strategy to overcome the defects of Ptx. Guo et al. [64] selected PEG2K-DSPE as the backbone to construct aptamer-functionalized shell core nanoparticles. Then, Ptx was wrapped in the core and three siRNAs were assembled by a calcium phosphate (CAP) shell (Ptx/siRNAsNPs-Apt) (β-Tubulin III, AR, and CXCR4), and the core was modified with a PSMA aptamer. Their experimental results showed that Ptx/siRNAsNPs-Apt can effectively encapsulate Ptx, improve the bioavailability of Ptx, and reduce its toxic and side effects. Collectively, the Ptx/siRNAsNPs-Apt showed an enhanced tumor-targeting ability, and achieved superior efficacy in the subcutaneous and orthotopic prostate cancer tumor model with minimal side effects.

### 2.7. Aptamer–Poly(amidoamine) Conjugates

Poly(amidoamine) (PAMAM) dendrimers are well-defined, highly branched macromolecules with numerous active amine groups on the surface [65]. Due to their unique properties, PAMAM dendrimers have steadily grown in popularity in drug delivery, gene therapy, medical imaging, and diagnostic application. EpDT3 is a 19-nt RNA aptamer, which can specifically bind to cancer cells overexpressing the epithelial cell adhesion molecule (EpCAM) on the cell surface and be endocytosed after binding with molecules [66,67]. Tai et al. [68] attached an EpDT3 aptamer and polyethylene glycol (PEG)-targeting EpCAM to the surface of the poly(amidoamine) (PAMAM) dendritic molecule, developed a new vector, PAMAM-PEG-EpDT3, and delivered the plasmid-encoded tumor suppressor plasmid-encoding lncRNA MEG3 (pMEG3) to CRPC cells. PC-3 and DU-145 cells were established as a CRPC cell model in vitro and a xenograft mouse model. The results showed that PAMAM-PEG-EpDT3/pMEG3 had good anticancer effects, as shown by the significant inhibition of the proliferation of CRPC cells. The inhibitory rate of the PAMAM-PEG-EpDT3/pMEG3 group was 63.34%, which was higher than the 40.95% rate of the PAMAM-PEG/pMEG3 group. The EpDT3 aptamer enabled the drug to accumulate at the tumor site, improved the transfection efficiency of pMEG3, and inhibited the proliferation of cancer cells. PAMAM-PEG-EpDT3 showed excellent CRPC cell-targeting ability. In vitro and in vivo models confirmed the anti-CRPC effect of PAMAM-PEG-EpDT3/pMEG3NPs, indicating its great potential as a gene therapy drug for patients with CRPC.

### 2.8. Aptamer–Thymoquinone Delivery Systems

Thymoquinone (TQ) is a natural compound known for its anticancer activity [69]. He et al. [70] generated planetary ball-milled nanoparticles (PBM-NPs) made with a natural polysaccharide, containing TQ, and coated with an RNA aptamer, A10, which binds to PSMA. The combination was used to explore the inhibitory effect of TQ in docetaxel-resistant C4-2B-R and LNCaP-R cells with the high expression of the hedgehog. They proved that PBM-NPs can specifically recognize the PSMA protein, bind to specific targets, and deliver drugs to target cells due to their negatively charged and small particles, and high binding ability to the positively charged membrane.

### 2.9. Aptamer–TPEN Delivery Systems

Scientists found that Zn^2+^ seems to promote citrate accumulation by inhibiting aconitase [71]. When the level of Zn^2+^ is significantly reduced, the ability of prostate cancer cells to accumulate citrate is reduced. This may induce the early response of prostate cancer [72]. Therefore, regulating Zn^2+^ may be a therapeutic strategy for the treatment of prostate cancer. The concentration of Zn^2+^ in prostate cancer tissue will decrease further with the progression of a malignant tumor. Supplementation of prostate cancer cells with Zn^2+^ can induce apoptosis and reduce the level of proangiogenic and metastatic cytokines, which indicates that reducing the concentration of Zn^2+^ is very important for the survival of prostate cancer cells. N, N, N′, N′-tetrakis (2-pyridylmethyl)-ethylenediamine (TPEN) is a Zn^2+^ chelator. TPEN can kill prostate cancer cells and lead to the imbalance of reactive oxygen species through an apoptotic mechanism [73]. SZTI01 is a DNA aptamer that can specifically bind to PSMA. Stuart et al. [74] evaluated the prostate cancer selectivity and therapeutic effects of liposomes (targeted and non-targeted) and free TPEN through in vitro experiments and in tumor-bearing mice. The results showed that the delivery of TPEN with aptamer-targeted liposomes can lead to specific delivery to targeted cells. In vivo experiments showed that aptamer-targeted liposomes loaded with TPEN reduced the growth of the tumor in a human prostate cancer xenotransplantation model. Taken together, the potential of the SZTI01 aptamer to selectively deliver the liposome TPEN to prostate cancer cells was proved to work as a potential new method for targeted treatment of CRPC.

## 3. Perspective

The development of aptamers has increased the potential success of the combination strategy of precise drug delivery systems. Aptamer–drug delivery systems have been used to explore aptamers, with the aim to significantly reduce the side effects of drugs, enhance the affinity to target cells and specific identification of target cells, and minimize the collateral damage to non-target cells. Some aptamer–drug delivery systems do not even damage normal cells, and thereby protect normal tissues and improve the patient’s prognosis. With the development of SELEX technology, dual aptamers have emerged that can simultaneously recognize two target cells. This technology will greatly improve drug administration efficiency. Another advantage of aptamer–drug delivery systems is that they can be easily controlled and quickly inactivated by antisense oligonucleotide “antidotes”.

Aptamer–drug delivery systems comprise aptamers and drugs with or without a carrier to explore the precise delivery of drugs. Aptamer–drug delivery systems have been developed into liposomes, micelles, nanoparticles, myristilated chitosan nanogels, super paramagnetic iron oxide nanoparticles, etc. Nanoparticles have become the main pharmaceutical preparation for aptamer–drug delivery systems because of their low molecular weight and easy absorption. Aptamer–drug delivery systems made into nanoparticles can improve the drug loading and the physical and chemical stability of the drug. They can also accelerate the drug’s release, enhance its absorption and cytotoxicity to target cells, and improve its efficacy. In addition, liposomes can reduce the toxicity of drugs and enhance the drug’s stability by encapsulating aptamer–drug delivery systems. However, the development of aptamer–drug delivery systems is limited to cell and animal experiments, and these systems have not been put into clinical practice. The transformation from experiment to clinical application is still a huge challenge. In addition, scientists should also consider factors such as the production cost, large-scale production technology, and patient acceptance.

In this study, we collected nine compounds (Figure 1) that were designed as aptamer–drug delivery systems for the treatment of prostate cancer (Figure 2). Dox is the most studied among these compounds. Scientists have used it to construct conjugates with different aptamers, and then to create different preparations. The results of this research suggest that aptamers can improve the efficacy of Dox and reduce its side effects. These achievements suggest that aptamer–Dox delivery systems have a good clinical prospect against prostate cancer. There are few studies on the other eight compounds in this field (Table 2). Among these compounds, some are drugs for the clinical treatment of prostate cancer, some are too toxic to be used in the clinic, and some are natural products that have not been used in clinical treatment. These studies expand the breadth of aptamer–drug delivery systems, suggesting that scientists can develop a variety of compounds for aptamer–drug delivery systems since this mode of administration can overcome the original defects of many compounds. These studies also confirmed the positive role of aptamers in the process against prostate cancer, thus promoting the application of aptamers in the development of prostate cancer drugs.

Here, we focus on summarizing the recent research progress of aptamer–drug delivery systems against prostate cancer. Scientists and clinicians continue to explore new therapeutic methods for the treatment of prostate cancer. Minimally invasive surgery, hormone therapy, radioimmunotherapy, vaccines, ADCs, photothermal therapy, gene therapy, etc., are currently playing, or are ready to play, a role in the clinic. However, none of these treatments can completely reverse the outcome of patients with advanced prostate cancer. Encouragingly, the aptamer may be a factor that can change this situation. Here, we propose a scientific assumption (Figure 3). Firstly, prostate cancer cells need to be isolated from patients through minimally invasive technology for in vitro culture. Secondly, cell SELEX technology can be used to screen aptamers that can specifically identify patients’ prostate cancer cells. Thirdly, the aptamers and cytotoxic drugs are prepared as conjugates and administered to achieve specific cell death in patients with prostate cancer. Through this administration mode, aptamer–drug delivery systems can precisely identify tumor cells in situ or tumor cells with distant metastasis, and then kill all tumor cells without damaging normal cells and tissues. Another advantage of aptamer–drug delivery systems is that individualized precision treatment can be achieved in this way to address individual differences in prostate cancer patients. However, this is an ideal scientific hypothesis. To realize this idea, many technological questions need to be addressed, such as how to quickly and safely obtain tumor cells from patients, how to quickly screen and synthesize target aptamers, and how to quickly synthesize aptamer–drug delivery systems. However, we believe that science will change the future and aptamer–drug delivery systems will shed light on the precise treatment of prostate cancer.

## Figures and Tables

**Figure 1 molecules-27-03446-f001:**
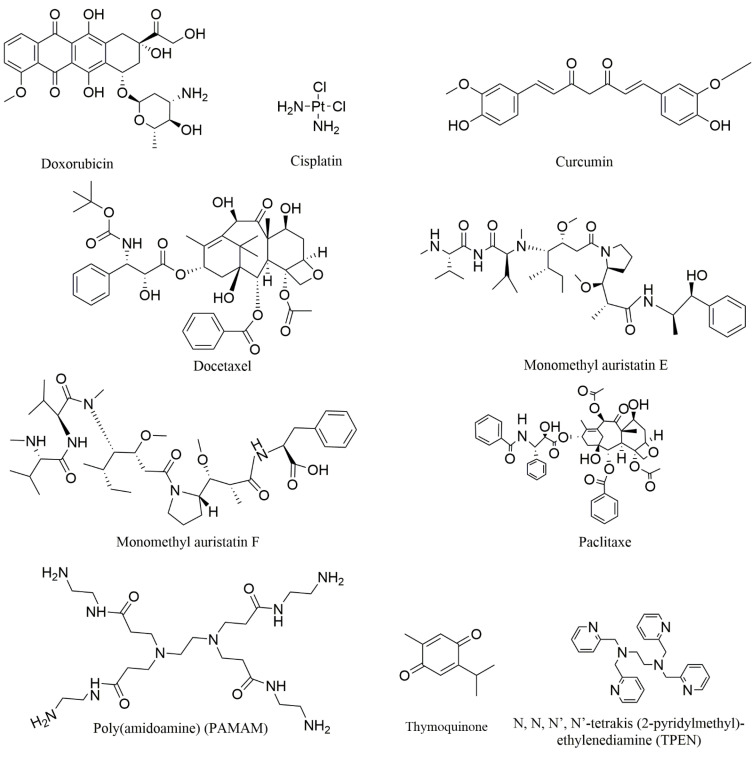
The chemical structures of the mentioned compounds.

**Figure 2 molecules-27-03446-f002:**
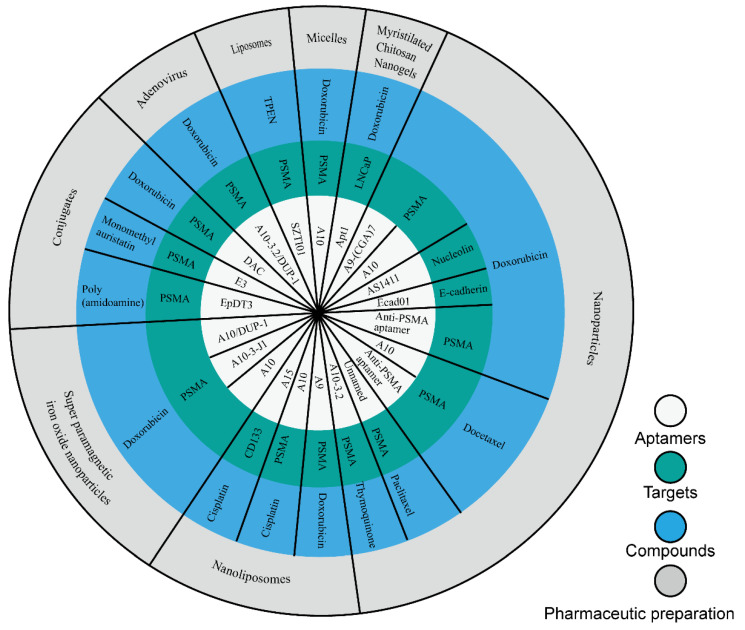
Aptamer–drug delivery systems for the treatment of prostate cancer.

**Figure 3 molecules-27-03446-f003:**
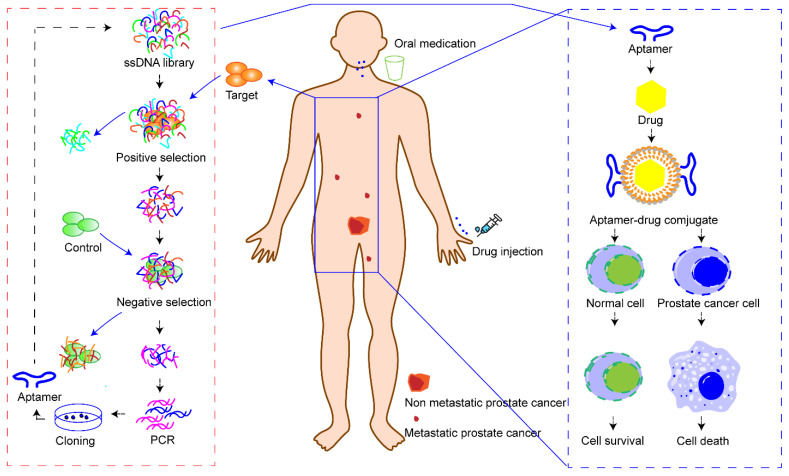
Overview of the action mode of aptamer–drug delivery systems. Firstly, tumor cells are ex-tracted and isolated from patients. Secondly, patient’s cancer cells are used as targets to screen aptamers that can specifically recognize the targets. Thirdly, a precise drug delivery system is constructed with a specific aptamer and drug to treat patients.

**Table 1 molecules-27-03446-t001:** Information regarding aptamer–doxorubicin delivery systems.

No.	Aptamer Code	Type of Aptamer	Base Sequence (5′-3′)	Target Protein	Target Cell	Drug	Pharmaceutical Preparation	References
1	DAC	DNA	dCGGCA_16_GCCG or dCGGCT_16_GCCG	PSMA	C4-2	Doxorubicin	Conjugates	[20]
2	A10-3.2/DUP-1	RNA/Peptide	A10-3.2: the 3′-end was modified with the amino “5′-GGGAGGACGAUGCGGAUCAGCCAUGUUU ACGUCACUCCU-(CH_2_)_6_-NH_2_-3′(with 2′-fluoro pyrimidine modifications)”, and the 5′-end was labeled with fluorescein isothiocyanate (FITC). DUP-1: N’-FITC-FRPNRAQDYNTN	PSMA	LNCaP, PC-3	Doxorubicin	Adenovirus	[22]
3	A10	DNA	GGGAGGACGAUGCGGAUCAGCCAUGUUUACGUCACUCCUUGUCAAUCCUCAUCGGC	PSMA	CWR22Rv1	Doxorubicin	Micelle	[25]
4	Apt 1	DNA	CATCCATGGGAATTCGTCGACCCTGCAGGCATGCAAGCTTTCCCTATAGTGAGTCGTATTACGAGCTCGAGCCTAGGCAG		LNCaP	Doxorubicin	Myristilated Chitosan Nanogel	[29]
5	A9	RNA	GGGAGGACGAUGCGGACCGAAAAAGACCUGACUUCUAUACUAAGUCUACGUUCCCAGACGACUCGCCCGACGA	PSMA	LNCaP	Doxorubicin	Nanoliposomes	[23,24]
6	A9-(CGA)_7_	RNA	GGGAGGACGAUGCGGACCGAAAAAGACCUGACUUCUAUACUAAGUCUACGUUCCCAGACGACUCGCCCGACGACGACGACGACGACGACGACGA	PSMA	LNCaP	Doxorubicin	Nanoparticles	[30]
7	A9-(CGA)_7_	RNA-DNA	GGGAGGACGAUGCGGACCGAAAAAGACCUGACUUCUAUACUAAGUCUACGUUCCCAGACGACUCGCCCGACGACGACGACGACGACGACGACGA	PSMA	LNCaP, 22RV1, DU145	Doxorubicin	Nanoparticles	[31]
8	A10	RNA	GGGAGGAcGAuGcGGAucAGccAuGuuuAcGucAcuccuuGucAAuccucAucGGc (3′-3′dT)-5′)	PSMA	LNCaP	Doxorubicin	Nanoparticles	[26]
9	anti-PSMA aptamer	RNA	NH2-spacer-GGGAGGACGAUGCGGAUCAGCCAUGUUUACGUCACUCCUUGUC-AAUCCUCAUCGGC invertedT-30 with 20-fluoro pyrimidines, 30-inverted T cap, and 50-amino group attached by a hexaethyleneglycol spacer	PSMA	LNCaP	Doxorubicin	Nanoparticles	[32]
10	AS1411	DNA	Apt1: TATGGTGAAGGGAAAGGTGGTGGTGGTTGTGGTGGTGGTGGAAACACCAAACCCAA Apt2: TTGGGTTTGGTGAAAGGTGGTGGTGGTTGTGGTGGTGGTGGAAACCTCCTTTCCTT Apt3: AAGGAAAGGAGGAAAGGTGGTGGTGGTTGTGGTGGTGGTGGAAACCCTTCACCATA	Nucleolin	PC-3, 4T1	Doxorubicin	Nanoparticles	[34]
11	Ecad01	DNA	GTGGGCTCAAGAAGAAGCGCAA	E-cadherin	DU145	Doxorubicin	Conjugates, Nanoparticles	[35]
12	A10	RNA	GGGAGGACGAUGCGGAUCAGCCAUGUUUACGUCACUCCUUGUCAAUCCUCAUCGGC	PSMA	LNCaP	Doxorubicin	Quantum dots, Nanoparticles	[28]
13	A10-3-J1	RNA-DNA	GGGAGGAAUAGCUGACGGGAGGACGAUGCGGAUCAGCCAUGUUUACGU CACUCCUUGUCAAUAAUAAGGGGC	PSMA	LNCaP	Doxorubicin	Super paramagnetic iron oxide nanoparticle	[36]
14	A10/DUP-1	RNA/Peptide	A10: TAATACGACTCACTATAGGGGAGGACGATGCGGATCAGCCATGTTTACGTCACTCC TTGTCAATCCTCATCGGC DUP-1: N′ biotin-FRPNRAQDYNTN	PSMA	LNCaP, PC-3	Doxorubicin	Super paramagnetic iron oxide nanoparticles	[37]
15	A10	RNA	GGGAGGACGAUGCGGAUCAGCCAUGUUUACGUCACUCCUUGUCAAUCCUCAUCGGC	PSMA	LNCaP	Doxorubicin	Super paramagnetic iron oxide nanoparticles	[38]

**Table 2 molecules-27-03446-t002:** Information of aptamer–drug delivery systems.

No.	Aptamer Code	Type of Aptamer	Base Sequence (5′-3′)	Target Protein	Target Cell	Drug	Pharmaceutical Preparation	References
1	A10	RNA	5′-NH2-spacer GGGAGGACGAUGCGGAUCAGCCAUGUUUACGUCACUCCUUGUCAAUCCUCAUCGGCiT-3′ containing 2′-fluoro pyrimidines, a 3′-inverted T cap, and a 5′-amino group attached by a hexaethyleneglycol spacer	PSMA	LNCaP	Cisplatin	Nanoliposomes	[44]
2	A15	RNA	SH-CCCUCCUACAUAGGG	CD133	DU145	Curcumin	Nanoliposomes	[51]
3	A10	RNA	GGGAGGACGAUGCGGAUCAGCCAUGUUUACGUCACUCCUUGUCAAUCCUCAUCGGC	PSMA	LNCaP	Docetaxel	Nanoparticles	[55]
4	Anti-PSMA aptamer	RNA	5′-NH2-GGGAGGACGAUGCGGAUCAGCCAUGUUUACGUCACUCCU (CH2)6-S-S-(CH2)6-OH-3′ with 2′-fluoro pyrimidines	PSMA	LNCaP	Docetaxel	Nanoparticles	[56]
5	E3	RNA	GGCUUUCGGGCUUUCGGCAACAUCAGCCCCUCAGCC	PSMA	22Rv1	Monomethyl auristatin	Conjugates	[59,60]
6	Unnamed	RNA	5′-NH_2_ (CH_2_)_6_ GGGAGGACGAUGCGGAUCAGCCAUGUUUACGUCACUCCUUGUCAAUCCU- CAUCGGCiT-3′ with 2-fluoro pyrimidines, a 3-inverted T cap	PSMA	LNCaP	Paclitaxel	Nanoparticles	[64]
7	EpDT3	RNA	5ʹ/5thiol/-GCGACUGGUUACCCGGUCG-3′	EpCAM	PC-3, DU-145	Poly(amidoamine)	Conjugates	[68]
8	A10-3.2	RNA	GGGAGGACGAUGCGGAUCAGCC AUGUUUACGUCACUCCU-spacer-NH2	PSMA	C4-2B-R, LNCaP-R	Thymoquinone	Nanoparticles	[70]
9	SZTI01	DNA	dGCGTTTTCGCTTTTGCGTTTTGGGTCATCTGCTTACGATAGCAATGCT	PSMA	C4–2	TPEN	Liposomes	[74]

## Data Availability

Not applicable.

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
