# Peer review of "Insights into Aptamer–Drug Delivery Systems against Prostate Cancer"

_molecules, 2022, doi:10.3390/molecules27113446_

Round 1

Reviewer 1 Report

The manuscript entitled "Insights into aptamer-drug conjugates against prostate cancer" submitted by Wang and collaborators reviews recent advances of different aptamer-drug conjugates to treat prostate cancer cells.

-This manuscript is hard to read and is presented as a catalogue of different Apt-drug conjugates. The authors should improve their writing.

- They should precise in the text the targeted receptors by the aptamer.

- When they start their writing by Author et al (Example Boyacioglu et al), they should immediately incorporate the reference such as Boyacioglu et al14.

- The authors should discuss which of the described ApDC is promising to go further for clinical use.

In conclusion, the authors should deeply improve their manuscript before considering its publication in Molecules journal.

Author Response

Dear reviewer,

Thank you for your comments concerning our manuscript entitled “Insights into aptamer-drug delivery systems against prostate cancer”. (ID: molecules-1696940). Those comments are all valuable and very helpful for us to revise and improve our paper. We studied comments carefully and have made related corrections in the revised manuscript.

The responses to the comments are listed in the response file.

Reviewer 2 Report

This manuscript by Wang and coworkers is a potentially nice overview focused on aptamer-drug conjugates against prostate cancer.

After a brief introduction on prostate cancer, the authors generally described both antibody-drug conjugates, aptamer-drug conjugates and then focused on aptamers.

As a general comment, the topic is attractive and certainly of interest for the typical readership of Molecules. The target of this manuscript is certainly of great significance and is fully inserted into the promising research field of aptamer-based drugs for anticancer therapy.

However, the paper lacks clarity in the present form. Often it is not easy to read for the presence of small typos and grammatical mistakes which need to be fixed before the publication. Some parts are not logically organized, and this makes reading very difficult and sometimes incomprehensible.

In addition, some parts have to be enriched and better discussed.

In addition, I fear that the reference list is scarce. There are generally few references for a review (only 63) and many of them are old or not appropriate. Surely it is necessary to update and enrich the list of references, above all in the introduction, inserting more recent ones and carefully checking the inconsistencies in the reported references.

Below, I will give some suggestions, but they are not exhaustive.

A major shortcoming of the presentation is the lack of real data. In most cases, cytotoxicity, uptake, amounts, effects, etc. are not given in quantitative values (see examples below).

Thus, I suggested the publication on Molecules only after major revisions and updates of references.

Major points:

  1. There are liposomes, solid lipid nanoparticles, micelles, gold nanoparticles, and polymeric nanoparticles 2.

Reference 2, i.e. “Li, L.; Wang, W.; Xu, X.; Wang, H.; Liao, S.; Li, W.; Zhang, W.; Liu, D.; Cao, B.; Wang, S.; Shen, K.; Ma, D. Aptamer-based radioimmunotherapy: the feasibility and prospect in cancer therapy. Journal of Radioanalytical and Nuclear Chemistry  2011, 290 (2), 453-457” is old and not appropriate.

You can replace it with “Recent Advances in Nanocarrier-Assisted Therapeutics Delivery Systems, 2020, doi: 10.3390/pharmaceutics12090837” which is recent and more suitable.

However, a brief discussion of the main features of these nanosystems should be inserted for a less expert reader.

  1. Discussion of aptamers (page 2, lines 64-74).

This part seems to be strictly related to oligonucleotide-based aptamers and few features are described from peptide-based aptamers. For example, the SELEX procedure is a typically explored procedure to identify high affinity and selective oligonucleotide-based aptamers. But, what about peptide-based aptamers??? Also, pay attention to the acronym of SELEX which does not mean “"phylogenetiony of exponential enrichment”. However, the overall manuscript only describes the use of oligonucleotide-based aptamers, as DNA, RNA or chimeric DNA/RNA forms. So at the end of manuscript, I do not consider really useful the discussion (not exhaustive, among the other things) of peptide-based aptamers.

  1. Concerning reference 12 related to the SELEX process, this citation is related to aptamers in general and not to the SELEX process.

In order to provide suitable tools to the readers, who eventually want to deepen the SELEX strategy, more suitable references must be included. For example, you can insert recent papers such as:

- Advances in the application of modified nucleotides in SELEX technology. 2018, do: 10.1134/S0006297918100024

- Recent developments in cell-SELEX technology for aptamer selection, 2018, doi: 10.1016/j.bbagen.2018.07.029;

- SELEX methods on the road to protein targeting with nucleic acid aptamers, 2018, doi: 10.1016/j.biochi.2018.09.001;

- Aptamers: The "evolution" of SELEX, 2016, doi: 10.1016/j.ymeth.2016.04.020.

This is a selected list. The authors could refer in general to these recently published reviews, so to include all the aspects of a SELEX process.

  1. Boyacioglu et al invented a new dimeric aptamer complexes called DAC”.

The importance of dimeric aptamers should be underlined since it is a widely explored strategy to obtain better performing aptamers. You can generally refer to a recently published review on this topic: Riccardi, C. et al. Dimeric and Multimeric DNA Aptamers for Highly Effective Protein Recognition Molecules 2020, 25(22), 5227; doi: 10.3390/molecules25225227

  1. Table 1 is particularly useful to give a clear and immediate picture on the data discussed throughout the manuscript. However, I would suggest enriching it with oligonucleotide or peptide sequences or specifying them each time in the text.

  1. They proved that the aptamosome was significantly more cytotoxic towered targeted LNCaP cells than the nontargeted PC3 and A549 cells, suggesting that the aptamosome improved the targeting ability of Dox 16, 17

Quantitative data have to be provided for “more cytotoxic” statement.

Some page, line 124: thereby leading to a significantly higher cytotoxicity.

Provide quantitative values for cytotoxicity and particularly IC50 values, if available. These are two examples, but this problem is frequent throughout the manuscript. Please, address these issues.

  1. For each aptamer described, a suitable reference has to be provided for a deeper knowledge on it. For example, in the case of AS1411, you can cite a good overview on this aptamer that is:

Bates, P. et al. G-quadruplex oligonucleotide AS1411 as a cancer-targeting agent: Uses and mechanisms. Biochim Biophys Acta Gen Subj. 2017, 1861(5 Pt B):1414-1428. doi: 10.1016/j.bbagen.2016.12.015.

  1. I have remained a little bit disappointed during the lecture. You present your review article as a description of aptamer-drug conjugates since the proposed title. However, most of the described systems are not purely a conjugation (covalent or not) of an aptamer and a selected drug but they are based on nanoparticle-based drug delivery system in which the aptamer and the drug are embedded.

So, I will change the title so to highlight this aspect of your manuscript. In addition, I will specify that the simultaneous use of an aptamer (with specific therapeutic or targeting action) and an anticancer drug is intended to obtain synergic effects on prostate cancer cells and this purpose can be realized through their incorporation onto suitable nanosystems (beyond the direct chemical conjugation). With this statement or a similar one, you can introduce this topic.

  1. Aptamer-cisplatin conjugate: please use the plural conjugates and check it throughout the manuscript since this error is frequent.

In this context, also the paper of Shanta Dhar and colleagues is of relevance and has to be inserted (Targeted delivery of a cisplatin prodrug for safer and more effective prostate cancer therapy in vivo

Proc Natl Acad Sci U S A. 2011 Feb 1;108(5):1850-5. doi: 10.1073/pnas.1011379108).

In addition, besides cisplatin and platinum-based drugs, also ruthenium and other metal complexes have been directly (by covalent conjugation) or indirectly (via the use of a suitable platform) linked to specific aptamers.

For example, a recent paper should be cited and discussed: Luke K McKenzie et al. RSC Chem Biol. 2021 Nov 2;3(1):85-95. doi: 10.1039/d1cb00146a.

  1. Conclusions section should be conclusion and perspective, that is the authors should provide outlines and future hope for the use of oligonucleotides in prostate cancer disease.

Definitively, this section should be more impressive, offering a critical point on the state-of art as well as possible future developments in this research field.

  1. The addition of nice, resuming Figures on the described systems will be appreciated.

  1. This paper has also to be included:

Bethany Powell Gray et al. Tunable cytotoxic aptamer-drug conjugates for the treatment of prostate cancer. Proc Natl Acad Sci U S A. 2018 May 1;115(18):4761-4766. doi: 10.1073/pnas.1717705115

Minor points:

  1. Page 1, line 22: “…therapies of prostate cancer…”.

“Therapies for prostate cancer”… sounds better

  1. Page 1, lines 39-40: “At present, the drug therapies for prostate cancer are mainly endocrine therapy and chemotherapy.”.

Please, provide information on endocrine therapy.

  1. Page 1, lines 46-47: “many precise targeting technologies have been developing...”.

Please, correct it as “many precise targeting technologies have been developed…”

  1. Page 2, lines 51-52: “ADC is another rapidly developing drug delivery systems in recent years.”.

Fix the agreement mistake in this sentence. The subject is a plural noun, i.e. antibody-drug conjugates. So the use of the verbal form “is” is incorrect.

  1. Page 2, line 57: “PSMA-targeted ADCs.”.

Please, define PSMA.

  1. Page 2, line 57: “single strand.”.

Please, change with single-stranded.

  1. Page 2, lines 82-83: “This combined treatment method can enhance cytotoxicity, reduce IC50 and improve therapeutic effect to tumor cells but reduce damages to normal cells and tissues.”.

This sentence can be better rephrased as: “This combined treatment method can enhance cytotoxicity, reduce IC50 and improve the therapeutic effect on tumor cells reducing damages to normal cells and tissues.”

  1. Page 2, lines 90-92: “However, these anti-cancer drugs have various side effects, such as allergic reaction, heart injury, hair loss, bone marrow suppression, vomiting and bladder stimulation, which limits its clinical application.”.

As before, if the subject is anticancer drugs, you should use “their clinical application” and not “its clinical application”. Please, correct it.

  1. Page 2, line 92: “Boyacioglu et al invented a new dimeric aptamer complexes...”.

The term “invented” is not appropriate in this context. Please, replace it with proposed, designed or similar. The same expression is used after on page 4: “A DNA·RNA chimeric hybridization aptamer Apt·dONT-DEN was invented…” Please, change this sentence. In addition, a new is a singular form. Complexes is a plural noun. Please, change it.

  1. Page 3, line 109: “These finding suggested”.

Findings.

This is just to mention some examples of the typos and grammatical mistakes present in the manuscript. The authors should put more attention on writing and the English language needs to be improved by a native English writer.

Author Response

(The authors gave the same response as above.)

Reviewer 3 Report

In this review entitled “Insights into aptamer drug-conjugates against prostate cancer”, Wang et al focused on use of aptamer drug-conjugates (ApDC) in prostate cancer, summarizing the the published data on this topic and analyzing  these conjugates in terms of targets, delivery strategies and conjugated toxins. The topic is novel and interesting; with limited reviews already published in this field. The structure of the review is quite clear and fluent; so, in my opinion, in order to be suitable for publication, the original draft should be subjected only to some minor revisions.

Specific comments:

  1. In the lane 207 the sentence “Most of the aptamer target PSMA protein, and only one of them targets E-cadherin protein” is uncorrected because also nucleolin is involved as target.

  1. In the lane 214, there is a contradiction about cisplatin role in pancreatic cancer treatment in the two sentences “it is not the first-line drug for prostate cancer chemotherapy “ and “Cisplatin is the standard treatment for prostate cancer”. Please clarify this point.

  1. There is confusion in Figure 2, Table 2 and text of lines 369-379. What are the 9 compounds to which authors referred in the text? Are they the subgroup conjugated to doxorubicin? Compounds of Table 2 are them conjugated to drugs different from doxorubicin? Please correct in Figure 2 A15 conjugated to curcumin, not cisplatin.

  1. Perspective section should be improved by adding a comparison between the use of ADC and ApDC, pointing out the advantages of the latter. Moreover, it would be useful for the readers focusing to the more promising compounds for in vivo activity among those discussed in the review.

  1. Finally, in Figure 2 hypothesis about screening of aptamers using in vitro culture of isolated patients’ tumor cells is not well represented. Please refine on it.

Author Response

(The authors gave the same response as above.)

Reviewer 4 Report

The authors review the aptamer's drug conjugates against prostate cancer. Overall the manuscript is well written. However, there are some issues that needed to be addressed.  

Abstract: please remove,, on the web of science,, line 28

Keywords: the authors may include drug conjugate as a keyword to increase  manuscript visibility

Introduction:  line 46         poly-meric  or polymeric?

                         Line 57, please explain the acronyms PSMA, CRPC, ( in text )

                         Line 75 SELEX  Systematic evolution of ligands by exponential enrichment  and SELEX should be outside the quotation marks

  1. 1 line 97 please explain DAC-D (more clearly … is it DAC in complex with Doxorubicin?)

        Line 107 please explain LNCaP and PC3 in text

Line 135, please elaborate in a few words on What a quantum dot is./

Line 147 please explain CGA in the text and revised the abbreviation list

Line 154 please explain      Apt·dONT-DEN          in text

Line 197 please explain DUP in text and revised the abbreviations list

Line 211 Micelle or micelle?

2.2    OK

2.3 ok

2.4 ok

2.5

line 259  MMAE can not be used by itself and need to be linked to a monoclonal antibody – perhaps this information should be stressed out.

2.6 ok

2.7 ok

2.8 ok

2.9 ok

  1.  

line 350 - ,, Some ApDC would not even damage normal cells, that can protect normal tissues and reduce the damage to the patients’’ – please rephrase.

Line 409   although Figure 2 is explained in the text please elaborate on its legend.

Author Response

(The authors gave the same response as above.)

Round 2

Reviewer 1 Report

The authors have improved their manuscript but there are some points to be clarified before its acceptance for publication.

These points are cited below:

  • It is not necessary to have separate tables (Tables 2 and 4) for the aptamer sequences since there is some redundancy. The authors should merge the 2 tables and harmonize references for the same sequence;
  • The sequence given in Table 4 is not exactly the same as that described in the cited reference 26. The sequence in Ref 26 is as follow:

A10 PSMA RNA Apt ((C6-NH2): GGGAGGAcGAuGcGGAucAGccAuGuuuAcGucAcuccuuGucAAuccucAucGGc (3′-3′dT)-5′)

The authors should check all the original sequences cited in their manuscript.

  • On page 3, lines 105 – 107, for the same sentence the authors cited 2 references (lines 105 and 106: Boyacioglu et al 20 proposed a new dimeric aptamer complexes called DAC. DAC has a good binding affinity and therapeutic efficacy as a dimeric aptamer complex 21. DAC can specifically PSMA). On line 107, there is no reference number associated to Boyacioglu et al (ref???) developed a complex DAC-Dox (DAC-D) with…..

Author Response

Dear reviewer,

Thank you for your comments concerning our manuscript entitled “Insights into aptamer-drug delivery systems against prostate cancer”. (ID: molecules-1696940). Those comments are all valuable and very helpful for us to improve our paper. We studied comments carefully and have made related corrections in the revised manuscript.

The responses to the comments are listed in the response file.

Reviewer 2 Report

In this revised version of their manuscript, the authors sensibly improved the paper taking into account all the Reviewer's suggestions.

In the present form, the paper is better focused and centered in the background context.

So I suggested its publication in Molecules.

As minor comments, please correct Shanta Dha with Shanta Dhar at page 7, line 248.

I would suggest  adding a Figure with the chemical structures of the mentioned drugs (e.g. doxorubicin, paclitaxel, cisplatin, curcumin etc...)

At page 10, line 392: "explored aptamers" is better than "apply aptamer".

Author Response

(The authors gave the same response as above.)
